# A Qualitative Exploration of Challenges and Opportunities for Dog Welfare in Ireland Post COVID-19, as Perceived by Dog Welfare Organisations

**DOI:** 10.3390/ani12233289

**Published:** 2022-11-25

**Authors:** Blain Murphy, Claire McKernan, Catherine Lawler, Patrica Reilly, Locksley L. McV. Messam, Daniel Collins, Sean M. Murray, Rob Doyle, Natascha Meunier, Aiden Maguire, Simon J. More

**Affiliations:** 1Institute for Global Food Security, Queens University Belfast, Belfast BT7 1NN, UK; 2Department of Agriculture, Food and the Marine, Agriculture House, Kildare Street, D02 WK12 Dublin, Ireland; 3UCD Veterinary Sciences Centre, University College Dublin Belfield, D04 V1W8 Dublin, Ireland; 4Centre for Veterinary Epidemiology and Risk Analysis, University College Dublin Belfield, D04 W6F6 Dublin, Ireland; 5Animal Health Ireland, Apartment 3, Carrick-On-Shannon, N41 WN27 Co. Leitrim, Ireland

**Keywords:** dog welfare, qualitative research, animal welfare, challenges and solutions, thematic analysis, COVID-19, dog welfare organisations, perceptions

## Abstract

**Simple Summary:**

This study was conducted to ascertain the challenges in dog welfare and the future needs of dog welfare organisations (DWOs) in Ireland, as perceived by the DWOs themselves. Minimal previous research into dog welfare in Ireland was identified. Using interviews and focus groups, this study aimed to highlight previously unknown challenges and opportunities, to improve future dog welfare. DWOs perceived a need for greater education of the public on dogs, increased enforcement of current legislation for welfare, action on dog breeding establishments and puppy farms, and the financial challenges in dog welfare. The study also identified the current principles applied in dog rehoming procedures by DWOs and the desire amongst DWOs to improve standards across the sector. Greater communication between voluntary and statutory organisations and reducing volunteer burden were identified by DWOs as potential solutions. The findings, while consistent with research findings in other international jurisdictions, also provide greater depth and interrogation of the Irish context. In conclusion, this study is the first step in identifying the challenges faced by the sector and provides recommendations from those deeply involved in dog welfare.

**Abstract:**

This novel qualitative study identifies challenges and opportunities to improve dog welfare in Ireland, as perceived by dog welfare organisations (DWOs), a previously underutilised stakeholder. This study sought the views of this predominantly voluntary sector of the next steps for policy and action in dog welfare, in light of the effects of the “puppy pandemic”, increased costs and COVID-19 restrictions. An integrated online focus group and interview design involving DWOs was analysed using inductive thematic analysis. Thematic analysis identified 2 key themes: (1) Key challenges and solutions in general dog welfare and (2) Challenges and opportunities in the welfare organisation sector. DWOs perceived poor public awareness of appropriate dog-husbandry, inadequate legislation enforcement, negative impact of puppy farms, and increased financial and volunteer burden. DWOs helped construct a best practice rehoming pathway, identified how overall standards could be improved and recommendations to enhance dog welfare. The DWOs perceived an increased numbers of households acquiring dogs, difficulties in rehoming, and financial challenges as threatening their viability as organisations and Irish dog welfare. Greater enforcement of legislation, greater communication between organisations and the government, and more media awareness were seen as helpful by the DWOs to improve dog welfare standards and their organisations.

## 1. Introduction

The explosion in dog ownership due to the COVID-19 pandemic and consequent increased time spent working from home [1] has led to fears, for the next decade, in relation to mass abandonment of dogs due to high veterinary costs and the impact of the “puppy-pandemic” on dog welfare [2]. It has been reported anecdotally that dog ownership in Ireland has increased consistently over the last decade, with a marked increase during the COVID-19 pandemic [2]. However, due to multiple factors, these trends cannot be verified following a review of currently available data sources on the dog population in Ireland, conducted as part of a multi-study research programme of which this project is a part [3]. The lack of data and insight currently available on Irish dog welfare is the rationale behind the multi-study research programme; of which this study will help provide the context to the other findings and inform future policy development in Ireland. Information relating to dog ownership is not routinely collected in Ireland, with the annual dog license considered a poor proxy measure to estimate dog population size. While 201,146 licenses were issued in 2021 [4], the European pet food industry estimated that there were 457,000 dogs living in approximately 25% of Irish households in 2021 [5]. Research to provide a robust evidence base on the owned dog population of Ireland, for the benefit of both charity organisations and the Irish government, including challenges and opportunities in relation to dog ownership and welfare, was identified in the first component of the research programme [3]. The current study utilises a qualitative exploration of the perspectives of dog welfare organisations (DWOs) in Ireland in relation to the challenges that charities are experiencing and signposts the need for changes in behaviour, ownership, and legislation in order to improve dog welfare in Ireland.

DWOs were selected as the study focus. The broad role of DWOs in Ireland includes receiving surrendered dogs from the public, rehoming dogs, educating prospective owners and children and providing care for unhealthy dogs. DWOs interact with multiple stakeholders in Irish dog welfare (such as prospective owners, local and national government officers, other welfare organisations and An Garda Síochána (Ireland’s national police and security service). Many DWOs receive governmental funding from the Department of Agriculture, Food and the Marine (DAFM). To be eligible, organisations must hold charity status via the Irish Charities Regulator. In total, EUR 13.9m was allocated to animal welfare organisations, including DWOs, between 2016 and 2020 [3]. Some DWOs may also operate as dog control shelters. In a literature search, minimal qualitative research has been conducted with DWOs. There have been some quantitative surveys and reviews conducted in the UK, which highlighted the large role of DWOs in educating the public on dog ownership and develop campaigns on welfare issues [6,7]. In a review of UK dog welfare priorities, DWOs/ charities in co-operation with local and national government have produced reports and guidelines on welfare; however DWO’s perspective on current and future challenges and solutions is a novel topic. Consequently, the practices and perspectives of these organisations are of national interest and an under-heard (from a research perspective) group whose contributions can add value to the discussion.

Dog ownership legislation in Ireland is predominantly established in The Control of Dogs Act 1986 [8], The Dog Breeding Establishments Act 2010 [9], Microchipping of Dogs Regulations 2015 [10] and Animal Health and Welfare (Sale or Supply of Pet Animals) Regulations 2019 [11]. These legislative instruments include requirements in relation to dog licensing, dog breeding, microchipping and sale and supply of pet animals. The role of each legislative instrument has been discussed in our previous work [3].

The International Partnership for Dogs (IPFD) has concluded that multi-stakeholder engagement is required to combat dog welfare challenges, including engagement between DWOs, kennel clubs, state services, veterinarians, and dog owners [12]. Additionally, the breed-specific strategies for health and breeding, supply and demand and behaviour were prominent themes in their report. Poor practices in some dog breeding establishments (DBEs) have been reported in the news media [13,14,15]. Whilst dog breeding is regulated in Ireland [9], the ethical issues surrounding large-scale dog breeding are controversial. An oversupply of dogs has been linked to poor dog welfare, negative ecological and public health impacts and is an economic burden for state bodies [16]. Poor planning and inadequate housing for dogs also poses a risk to public health and the environment [17]. An increasing demand for breeds susceptible to brachycephalism and other conditions, which negatively affect canine health, have been linked to social influence and exposure, rather than to functionality or suitability (due to health and trainability concerns) [18].

Dog welfare is a wide-ranging topic, with multiple challenges and varied legislation and practices. The research programme’s review study identified challenges in relation to dog sales, breeding, exports and imports and data quality for registration and identification of dogs [3]. Exploring these challenges and the roles of DWOs in the Irish welfare community formed the initial nucleus of our research aims.

## 2. Materials and Methods

### 2.1. Design

This study integrated online focus groups and interviews to identify and explore the experiences of DWOs. The findings will help develop a model of the current and future challenges for dog welfare and DWOs in Ireland, to provide data to inform future Department of Agriculture, Food and the Marine (DAFM) policies. A qualitative study design was selected due to the lack of research in relation to DWOs in Ireland and the desire to explore the previously under-researched experiences of these organisations in relation to dog welfare.

An integrated focus group and interview approach was used to increase participation and provide a wider range of opinions and debate [19]. This multi-method mono-strand design (combining focus groups and interviews) is a more holistic approach to data richness [20] and endorsed by a critical review [19] as leading “*to an enhanced description of the phenomenon’s structure and its essential characteristics*”. The first author has used this approach previously [21].

This study used a purposive sample of organisations in Ireland involved in dog welfare. Most of these were in receipt of animal welfare grant funding from the DAFM between 2019–2021 and all held charity status. National animal welfare policy is overseen by DAFM [22]. Charity-registered animal welfare organisations may apply for DAFM animal welfare grant funding to assist in delivery of animal care and animal welfare services [23]. An invitation to participate in this study was sent via email by the first author (BM) to 74 of these organisations. Two subsequent reminder emails were sent to non-respondents to encourage participation. Organisations were offered the choice of participation in a focus group or an individual interview. Participants were informed that their participation was anonymous outside of their specific focus group, and that they would be interacting with organisations representatives openly within their focus group session. Anyone with any issues with this lack of anonymity was offered an individual interview. Participation was managed by the first author to comply with the EU General Data Protection Regulation (GDPR) and ensure confidentiality separate from the DAFM. Organisations were informed that their participation in the focus group was not anonymous. To participate in this study, organisations selected were those that include dog welfare as part of, or their only, remit. This includes DWOs which hold dogs on their premises and co-ordinate fostering. Organisations were sampled to ensure a country-wide spread, based on their geographical location in the Republic of Ireland. Organisations based in Northern Ireland were not included. No remuneration was given to respondents for their time.

In total, 27 DWOs participated in three online focus groups (FGs) and 16 online interviews. All discussions were held on Zoom, due to COVID-19 restrictions and for participant safety. Three interviews were triadic interviews, with two representatives from each organisation present, due to the different skills and responsibilities each representative held in their respective DWOs. These triadic interviews were conducted in the same session/call, and equal weight was given to triadic and dyadic interviews in the overall analysis. Five organisations declined to speak after initial contact, due to poor online connectivity, lack of time or feeling they were not suitable candidates to participate in the study. As the individuals were representing their organisation rather than themselves, only organisational-level demographic data was collated. Basic demographics of the organisations are provided in Table 1. Focus groups and interviews were conducted between January and May 2022.

Both the interviews and focus groups followed the same semi-structured schedule developed by the study team in consultation with the published literature (Appendix A), with the aim of identifying the challenges and potential solutions in two domains, as perceived by participating DWOs: that of general dog welfare and the role of DWOs in Ireland. Focus groups lasted approximately 90 min, whilst the interviews lasted approximately 60 min. Both were led by an experienced male qualitative researcher (BM). Focus groups and interviews were recorded on Zoom, transcribed verbatim, and checked against interview recordings and relevant field notes. Transcription was conducted by an external agency and the transcripts were checked by BM and CMcK for precision. Recruitment was carried out until a point of saturation, meaning that “sufficient data to account for all aspects of the phenomenon were obtained” [24]. Draft transcripts (one-page extracts) were sent to a subset of organisations for respondent validation. The Consolidated Criteria for Reporting Qualitative Studies (COREQ) [25] guided the reporting of the study. Ethical approval was obtained from University College Dublin’s (UCD) Research Ethics Committee (Research Ethics Reference Number LS-E-21-26-More).

### 2.2. Data Analysis

The interview and focus group transcripts were analysed using inductive thematic analysis [26] following the 6-step trustworthiness criteria [27], to establish trustworthiness and a procedural methodology for inductive thematic analysis. This involved: (1) achieving data familiarisation by repeatedly re-reading transcripts (2) generating initial codes, highlighting words/phrases relevant to the research questions; (3) Organising codes into subthemes; (4) arranging sub themes into key themes and (5) defining final key themes [27]. These data rounds were then discussed by the two researchers (BM and CMcK) independently and presented to members of the wider interdisciplinary research team for further analytical discussion. The iterative rounds continued until there was agreement that no new data, or no new themes, emerged from the transcripts.

Quotations are provided to support the findings and have been edited, with the use of “//” to identify where data/quotations were altered, for readability and grammar. The data were organised using NVIVO v12 [28]. Theme development was similar between the data collection methods.

## 3. Results

In total, 27 DWOs participated in the study (Focus group: *n* = 11 (FG1: *n* = 4, FG2: *n* = 4 & FG3: *n* = 3) and Interviews, *n* = 16). These organisations were geographically spread, with two-thirds of Ireland’s counties represented in the data. The age of these organisations (years since establishment) ranged from over 100 years old to less than 5 years. A mix of organisation sizes were included, classified via previous governmental funding and dog capacity; with small (*n* = 10), medium (*n* = 10) and large (*n* = 7) organisations included, Table 1. There were minimal differences in the findings between the focus groups and interviews; with similar opinions and impact on theme development from both methodologies.

Thematic analysis identified two key themes: Key challenges and solutions in general dog welfare and challenges and opportunities in the welfare organisation sector; from which 8 separate sub-themes were divided. The sub-themes are a reflection of the opinions and beliefs of the DWOs interviewed. These can be seen in Figure 1.

### 3.1. Key Challenges and Solutions in General Dog Welfare, as Perceived by Participating Dog Welfare Organisations

The first theme includes key areas identified by DWOs in relation to general dog welfare in Ireland. These include the need for education of both children and adults in the general public in relation to dog husbandry, the impact of “puppy farms” on the Irish dog market, the role and enforcement of current animal welfare legislation in relation to dogs, challenges in the veterinary sector, and the impact of awareness and mass and social media on dog welfare in Ireland.

#### 3.1.1. The Need for Education and Awareness

The first sub-theme was related to dog husbandry education/awareness among the general public, with new/potential dog owners and school children highlighted as priority groups. Additionally, the role of the news media and social media in raising awareness of, and promoting, dog welfare in Ireland. This sub-theme was prominently raised in all of the focus groups and in the majority of interviews.

Participating DWOs emphasised the need for early education and intervention in schools. Outreach and educational involvement activities, such as talks and facility tours, were viewed as useful activities to encourage heightening awareness of the needs of dogs demonstrating appropriate animal husbandry techniques and illustrating how inappropriate dog handling may lead to aggressive behaviours. Currently it is felt by the DWOs, that these initiatives were predominantly driven by individual interests of teachers rather than standardised curricular or prescribed school activities.


*“Welfare should be working with behaviour, people, and longer term, because you need to educate the population about behaviour and in schools or with the kids.//Because people tend to think they know lots about dog behaviour and, in our experience, people know very little.”*
(I1)

Factors such as a low level of knowledge about dog care, exacerbated by the influx of new owners as a result of COVID-19 and low availability of in-person dog training and shortage of trainers, were repeatedly stressed by DWOs. More specifically, the DWOs suggested that there was limited awareness among the general public of the long-term time (approximately 10–20 years) and financial commitment required for dog care, and the need for continuous preventative healthcare measures. Poor training practices were linked, by the DWOs, to high rates of behavioural challenges, such as aggressive behaviour, separation anxiety and poor toileting, in surrendered dogs and in the wider dog welfare arena. As a solution, some larger DWOs were able to conduct dog training classes as part of their rehoming policies; predominantly over Zoom/Skype. This scalable education initiative was expected to remain post-pandemic in these DWOs. Mandatory dog ownership courses, with reference to Zurich/Switzerland practices [29], were seen as another possible solution by DWOs.


*“We need something government will run, but well run by people interested in animal welfare. More centralised with an educational unit. I also think schools could do with upping their education on animal husbandry really in general.//I think it needs to be little and often in schools. Start (the education) at quite a young age.”*
(I5)

The news media and publishers were seen by them as supportive and constructive in developing a better image of rescue dogs. Many DWOs used social media to showcase dogs available for adoption, and to increase awareness of dogs in welfare organisations. However, many DWOs felt they did not have the required training or expertise to take advantage of this. Greater media awareness and promotion of responsible dog owning was a desire of the DWOs, and positive coverage was linked by DWOs to surges in fundraising, animal/dog welfare and volunteer recruitment. TV exposés such as “RTÉ Investigates: Greyhounds Running for Their Lives” [30] were highlighted as positively affecting rehoming applications, especially for greyhounds. 


*“The starting point would be the RTÉ (Irish national broadcaster) documentary that came out, three years ago, the exposé thing. Burned into your brain. That was a bit of a watermark, because there’s been other kind of smaller exposés and in newspaper articles and stuff, but that was something that hit the broader public consciousness. And there was just that huge reaction to that, and then combined with the figures of the amount of money that the racing industry receives from government sources. //They were able to say in that programme that they have gone through and done the math with the dogs that were unaccounted for basically, from the racing industry, and they could say that there were 6000 greyhounds being destroyed every year in Ireland from the racing industry.//(The racing industry) don’t report the dogs they have, they don’t register them the right way, they don’t keep track of all the pups that are born, they only keep track of all the litters.”*
(FG3)

#### 3.1.2. Long- and Short-Term Impact of Puppy Farms

Fuelled by demand, dog breeding establishments (DBEs) produced thousands of puppies during the pandemic. DWOs repeatedly expressed concerns in relation to the conditions in which these dogs were kept/raised. The participating DWOs received multiple reports of dirty cages and facilities, low levels of vaccination or healthcare, poor nutrition, inappropriate breeding (both cross- and inbreeding), and behavioural issues due to poor enrichment and socialisation. Whilst some licensed DBEs were perceived positively by DWOs, such as those which supported rarer breeds, many were viewed as “puppy farms,” with a greater emphasis on *“a means of making money”* (I2) than dog welfare. Some DBEs are licensed for 200–300 breeding bitches (in addition to the male dogs and puppy litters); in the opinion of some DWOs, such numbers were considered unmanageable from a welfare point of view. Concerns about the regulations adherence, that standards of inspections were inconsistent, legal penalties should be higher and that microchipping practices were poor were also raised by DWOs in relation to some licensed DBEs in Ireland. DWOs supported greater financial penalties, removal of the tax breaks for dog breeding, and more importantly more robust enforcement of the minimum standards in the area.


*“Profiteering. The legislation does need to be enforced, but how you can provide welfare if you have 200 dogs?//What you are supposed to do every day for dogs in terms of enrichment and all of that sort of stuff. But if you’ve got 200 dogs under one roof, it’s very difficult to do that. These large-scale breeders really are not going to do welfare on any level. From a behavioural point of view, they’re producing problems.”*
(I1)

The participating DWOs spoke of the lucrative but undersupplied market for dogs during COVID-19, which encouraged increased breeding of dogs, especially “designer breeds” (which were more expensive). While the cost of an average puppy was between EUR 1000 and EUR 2000, some breeds such as French Bulldogs, could cost up to EUR 10,000, anecdotally. Many potential new owners were unaware of the conditions in which their dogs were raised, with multiple accounts reported by DWOs of subterfuge as to the origin of the puppies and of meetings with the prospective pet owner being conducted in a location other than the DBE itself. The prohibition of dog sales on websites such as DoneDeal [31] was seen by the DWOs as beneficial in reducing this.

However, the long-term effects of the heavy influx of dogs, specifically puppies, onto the Irish market during the pandemic leading to over-saturation was identified by the DWOs as the greatest challenge both to them and to dog welfare in Ireland over the next five years. As the interviews/focus groups proceeded from January to May 2022, particularly in the latter months, there was a greater emphasis placed by DWOs on the surrender of many of these *”pandemic puppies”* (I6) as well as to the resultant strain upon the resources of the welfare organisations, with dog owners increasingly returning to work outside the home as the project continued.

#### 3.1.3. Following through on Current Legislation

Enforcement of current legislation was seen as a huge challenge for dog welfare by participating DWOs. Whilst the underlying legislation was perceived as adequate by DWOs, enforcement and interaction with authorities (An Garda Síochána and authorised officers) were thought to be potentially problematic. Some DWOs were of the view that support and engagement from An Garda Síochána was limited in some animal cruelty cases, and reported that many counties either having none or shared access, with other counties, to authorised officers. Additionally, financial penalties for animal cruelty cases, especially in relation to puppy farms, were seen as inadequate, with the financial benefits of DBEs considered extremely lucrative in comparison. The idea of a task force, with specific An Garda Síochána resources, for animal/dog cruelty cases was raised as a potential solution. In addition, more authorised officers, with specific training and powers, were advocated. Welfare challenges were raised by DWOs with respect to dogs held in custody (as a result of an owner’s court case). They reported that some dogs were retained for over 2 years during court proceedings in dog control shelters and cited a need for a better system in relation to their management in such cases.


*“Zero tolerance. Because treating people with kid gloves is ridiculous and then they go to court and they get a slap on the wrist and a fine of 200 quid to the ISPCA and it’s over, they don’t have to worry about it anymore. The day that somebody gets an actual custodial sentence in this country for animal cruelty. Because we have to come up to the UK standards.”*
(FG 1:3)

Microchipping, which has been mandatory for all dogs in Ireland since 2016, was raised as a specific issue by DWOs. They reported that many of the dogs surrendered or rescued by them either were not microchipped or the data were incorrect in the microchipping databases. Some DWOs estimated that less than 15% of surrendered/rescued dogs were microchipped, with only 50% of those registered correctly. These animals were predominantly stray animals, with surrendered dogs more likely than rescued dogs to have been microchipped. There are currently four non-linked microchip databases present in Ireland, and dogs adopted from DWOs need to be registered by the new owners which costs approximately EUR 20. This financial cost as well as the new owners’ lack of familiarity with the system was viewed as a barrier to correct registration. Lack of traceability was raised as an issue by DWOs, and a challenge for reclaiming lost dogs from dog control shelters. According to the DWOs, little proof of ownership is required when claiming a dog from these facilities. The DWOs also reported a lack of uniformity in the provision of microchips to dogs coming from dog control shelters, with some dog control shelters microchipping dogs and others not. However, it was acknowledged by the DWOs that more traceable microchips (specifically history of ownership) raised questions in relation to GDPR and data protection and requires more in-depth specific research.


*“It’s the microchip, it’s a bit loose, not working as well as I’d like to see it work. It’s a very important tool in terms of welfare and in terms of traceability and stolen dogs and all that stuff. It’s sort of not really doing what it should be doing.//We have about 60 to 70 percent of all dogs in the country registered on the three or four databases that we have. Probably 50 percent of those have erroneous registrations, they’re still registered to the breeder.”*
(I9)

#### 3.1.4. Veterinary Care and Costs

The high costs of veterinary care represented a significant portion of the budget for many participating DWOs. They highlighted high medical costs, especially for older dogs, which were considered less likely to be rehomed in the future, as a rationale for surrenders they received. Some rescues estimated costs of approximately EUR 150-EUR 300 per dog for vaccinations, microchipping, neutering, de-worming and feeding and reported that those costs could rise sharply if a dog has medical issues. The DWOs reported that costs have increased as access to cheaper medicine imported from Britain has become more difficult due to Brexit. Medication was reported as being more expensive in Ireland than the UK. The majority of DWOs stated that their annual governmental animal welfare grant funding was primarily spent on veterinary fees. Some DWOs employed in-centre vets and highlighted similar challenges regarding costs. Solutions such as subsidised or free neutering initiatives were suggested by DWOs as potential solutions to ease the burden.


*“Most of the money goes on veterinary fees and feeding. You have one dog, ready for homing? You start off with vaccinations at 50 Euro, 50 Euro for microchipping, registration 15. And then you to neuter them, 140 to 160 Euro. Okay that’s worming defleaing, feeding. And anything else that might go wrong and once you’ve done that; you’re looking at nearly 300 Euro per dog. There’s nobody in my situation getting anything on labour costs.//You can look for a donation //when you’re homing the dog for around 100 to 125 Euro?”*
(I4)

DWOs that had in-centre veterinary personnel, highlighted significant recruitment difficulties in the sector (for both vets and vet nurses). The shortage of qualified dog trainers/behaviour specialists, and a lack of accreditation in the dog training/behaviour specialist professions, was also seen as a risk factor by DWOs for poor quality in the dog and wider animal welfare sector.


*“I think there’s a huge issue in the veterinary sector with the number of vets and reception staff that’s going to inadvertently, in three years’ time or less, have a knock-on effect on animal welfare because the preventative procedures (will not be completed) because they’re not emergency, they’re not life threatening.”*
(I3)

Additionally, inappropriate breed mixing (such as *“Bull Dogs and King Charles Cavaliers”* (I2) and *“Malinois crossed with Lurchers or (German) Shepherds”* (I16)) of dogs during the pandemic was highlighted by DWOs as a challenge for veterinarians and DWOs in the future. DWOs foresee extensive demand on the services of their organisations and veterinarians to treat these disease-prone breeds (especially in terms of skin issues) for which there is little data on temperament and lineage/ancestry.


*“They’re crossing them, nearly everything except a spider nowadays. It’s just unbelievable. And the stranger the name, the better.//A Jog, a Maltipoo, Malshee and a Cockapoo. The behavioural issues that’s coming out in those dogs is serious. Because the Cockapoos seemed to have been crossed with the Red Cocker spaniel which has issues with guarding and temperament issues anyway.”*
(I7)

### 3.2. Challenges and Opportunities in the Welfare Organisation Sector, as Perceived by Participating Dog Welfare Organisations

This theme highlights the key areas that participating DWOs identified in relation to their role as organisations and challenges they currently face. This was exemplified by the process of adopting dogs from a welfare organisation, and the administrative work needed to do this appropriately in order to increase the chance of a successful adoption.

#### 3.2.1. Getting a Dog from a Shelter

The processes followed by DWOs for dogs arriving at DWOs and then being rehomed were broadly similar across organisations. Best practice for many was an initial isolation period and veterinary examination, followed by neutering and vaccination. Some DWOs engaged in additional extensive behavioural training and temperament checks in order to match dog temperament and breed to potential owner lifestyle. Dogs were then advertised for rehoming, predominantly online through social media. Applications were usually online, with a pronounced demand for *“fluffers”* (such as bichon-frises and cockapoos) and designer or fashionable breeds (such as bulldogs and pugs). These breeds were attracting up to 4000 applications during the initial lockdowns during the COVID-19 pandemic.


*“The importance of assessing dogs properly and being honest with the people you give dogs to, telling them the good things and telling them the dog’s issues and problems. We follow up with every dog we rehome. We make contact again with the person. We get back to see how they’re getting on, if they have a problem, we encourage them to include us in the problem solving. And we develop a good relationship ongoing, and it helps, because then people will recommend us as a very responsible organisation, and that we have good follow through, and good contact. When people get their first dog, some people know nothing about the care and welfare of dogs. And it gives them a safety valve when they feel they can contact, who they rehomed the dog from for advice and for help when they have the problems. And I think those kinds of things would help welfare organisations greatly and approve the welfare of the dogs they rehome.”*
(I16)

As a result of COVID-19, many participating DWOs required appointments for visitors to their establishments and greater usage of technology to reduce the burden on, and risk to, staff considerably. Development of stringent rehoming policies by many of the organisations, involving home-checks, applications, financial assessments (to ensure new owners could afford to maintain their dog) and multiple meetings prior to adoption, were all seen by DWOs, as means to substantially improve the ability of dogs to integrate into their new homes. Some DWOs have invested significant resources into these policies, including behavioural training programs, profiling dogs to owner’s lifestyle, post-adoption classes and check ins, which many found had reduced returns of dogs. However, these policies require finances and time to maintain, which is a recurring issue. While working from home was more amendable to dog ownership, DWOs believed a lack of emotional engagement and problems with separation anxiety and engagement were potential issues for the future, especially in more active breeds like collies.

#### 3.2.2. Exporting of Dogs

Exportation of dogs has been a release valve for Irish DWOs over the last 30 years. However, the participating DWOs reported that the effects of Brexit and COVID-19 have led to a reduction of the practice; with the British market more difficult to export to (due to export health certification requirements), and the land bridge (travel via Britain) to Scandinavia (a popular destination) similarly hampered. Most DWOs had specific centres abroad they linked with and have established long-term relationships for the exporting of dogs across Europe and Britain. Whilst the importation of dogs into Ireland does occur, few DWOs had extensive dealings in this area due to existing oversupplies in the Irish market. DWOs reported that currently, greyhounds, lurchers and collies are the breeds most commonly exported from Ireland by them. DWOs espoused a more positive cultural acceptance and attitude to dogs by the general public in other countries (such as Germany and Sweden) and noted that whilst Irish perceptions of dog welfare has improved, Ireland lags behind other European nations in terms of welfare.


*“I think Irish dogs are going to have possibly less European rescues to help them. I think it’s time that Ireland started cleaning up its own mess.”*
(I10)

Greyhounds (and other sighthounds such as lurchers) were highlighted by DWOs as a particularly difficult group for DWOs to rehome in Ireland, with historical beliefs and misconceptions about the breed (including their need for exercise) cited. DWOs made comparisons between the government funding their sector receives and that of Greyhound Racing Ireland, believed by respondents to be approximately EUR 19 million per annum. A perceived lack of co-operation from Greyhound Racing Ireland was consistently highlighted by participating DWOs, especially regarding dogs which were raised to race but did not subsequently do so. Many of the DWOs rescue greyhounds directly from the industry (there are multiple greyhound-only welfare organisations), and most greyhounds needed to be exported to mainland Europe and Scandinavia due to the difficulties in finding homes for them in Ireland. One of the difficulties reported by DWOs was that greyhounds were not covered by the Animal Health and Welfare Act [11] (this perception, which was common in greyhound-related discussions, is not correct. Greyhounds are covered by the Animal Health and Welfare Act).


*“It’s not easy to take dogs from Ireland, or not as easy as it was. Our dogs going to Italy, our greyhounds going to Italy and a few lurchers, that’s unchanged. And obviously the dogs going to the UK now is significantly more challenging, because of Brexit. And if we didn’t need to, we wouldn’t send dogs to the UK anymore, because it is a paperwork nightmare.”*
(FG3)

#### 3.2.3. Volunteer Burden

The levels of administration and work needed for DWOs to function relied heavily on volunteers, with most organisations run exclusively by volunteers (some larger rescues had paid staff). However, participating DWOs reported that many volunteers see themselves as fulltime employees, but work as volunteers due to limited finances in the sector. Concern was expressed in relation to burnout amongst core volunteers and the retirement of older volunteers, without whom rescues could not function. Anecdotally, at least one well-established welfare organisation closed pre-pandemic due to retirements. As conventional fundraising proved difficult during the COVID-19 pandemic, many DWOs highlighted shortfalls in their revenue; however, new streams of online income were realised for some, for example, via public charity challenges and GoFundMe.

DWOs highlighted multiple administrative burdens on volunteers, including maintaining their charity status, keeping animal/veterinary records up to date, microchipping, making tax returns and applying for animal welfare grant funding. Some larger organisations advocated regionalised and larger centres of excellence for animal welfare. However, it was recognised by DWOs that the rural nature of Ireland and the volunteer-focus posed challenges to this model.

The participating DWOs identified few opportunities to collaborate and share their expertise. While many collaborated on home checks, greater co-operation was desired by some such as an annual conferences or training events. The benefits of a formalised training route (with certification) were also highlighted by DWOs, via online courses or community level courses. However, high emotionality in the sector was also seen as a challenge by some volunteers with personality clashes between volunteers/organisations described. Some DWOs reported that dogs with issues remain as long-term residents of rescues, and felt that this “hoarding” should be discouraged.


*“No two days are the same and no two animals are the same.//It’s very challenging to achieve everything.//The more funding you have, the more you can do, quite frankly. If we’re relying upon, which we’re very grateful for, but also more so we’re relying on donations and legacies and fundraising efforts. And it’s hugely challenging to keep on top of all of that //. There’s always a need for more space, there’s never ever enough space. It’s a very challenging work and it’s not for everybody.”*
(I8)

#### 3.2.4. Setting the Standards for Animal Welfare Organisations

Whilst there are minimum standards prescribed, many of the participating DWOs highlighted substantial inconsistencies in enforcement of standards. Whilst DWOs received annual inspections by a governmental inspector, other dog holding facilities (dog control shelters, DBEs and commercial kennels) were perceived by DWOs as not receiving similar levels of scrutiny, at county and country jurisdictional levels. Dog control shelters, dog licenses and inspections are predominantly administrated at county rather than national level, which was raised as an issue by DWOs. Many DWOs see inconsistencies between counties as a result of this. DWOs perceived themselves as having to fulfill greater standards than many of the puppy farms that their dogs originated from. Further, some DWOs raised concerns regarding the conduct and standards in some dog control shelters; citing cold and wet facilities, a lack of enrichment for the dogs and use of inappropriate equipment. The DWOs also reported on a need for more isolation kennels and better cleaning facilities.

In terms of standards, several DWOs highlighted standards set by the Association of Dog and Cat Homes (ADCH) (UK) and suggested that this template could be utilised to promote improvements in animal welfare in Ireland. It was acknowledged by the DWOs that requirements for facility standards may lead to closure of smaller welfare organisations, especially in rural areas. Further, the financial cost in establishing the correct facilities and procedures may be difficult for many DWOs which are currently struggling to meet their financial obligations. The DWOs advised that significant funding streams may be needed for capital projects to address this.


*“If I was advising the Irish government, I’d tell them to contact the ADCH. The regulation of rescues is coming in the UK, and the UK government are using the ADCH minimum standards and as part of their own standards. It’s the ADCH standards have been designed by the members and the members are ranged from foster based groups that don’t even have a centre //and the large heavy hitters like Battersea, RSPCA, Dogs Trust.//It’s based on the amount of dogs, ratio of Dog to animal, what’s your facilities are like, minimum size and space for each animal, a minimum amount of exercise, all that kind of stuff in terms and they are assessed, and even members are regularly externally assessed well before COVID.”*
(FG2:3)

## 4. Discussion

This novel study explores the experiences and insights of DWOs engaged in dog welfare in Ireland. Participating DWOs identified multiple challenges and opportunities to develop initiatives and improve practices in relation to dog welfare. The study found that DWO had a perception of poor owner awareness of dog welfare and husbandry, inconsistent enforcement of legislation, the negative impact of poorly operated puppy farms and the burden on volunteers. These were key pillars of DWO’s experiences. Additionally, this study highlights the financial strain on DWOs due to COVID-19 and their concerns for the future due to increasing costs arising from Brexit and the increase in the general cost of living. This study provides an academically rigorous foundation to help inform policy in relation to DWO’s perception of dog welfare in Ireland and provide recommendations for further research.

The most prominent and consistent topic of concern to Irish DWOs was their experience of a lack of understanding of appropriate dog husbandry; from educational, general public and pet owner perspectives. Additionally, concerns in relation to preventive healthcare (such as vaccination), dog training, financial planning (such as veterinary fees and insurance) and legal compliance (such as microchipping) were consistently highlighted by welfare organisations. This is consistent with previous research in Ireland, which found poor awareness among dog owners and non-dog owners that the responsibilities of dog ownership are legal requirements [32]. Specific issues in relation to a lack of understanding of dog identification, straying dogs, abandonment and health safeguarding requirements similarly corresponded to the previous study [32]. Developing better practices for dog welfare is a long-term initiative for DWOs and the Irish Government, under Ireland’s animal welfare strategy [22]. Some countries, such as Germany and Sweden, have introduced legislation guiding day-to-day handling of dogs; including minimum requirements for dog walking (sufficient exercise) and contact with a caregiver [32].

Many of the participating welfare organisations highlighted their belief that current legislation was suitably and sufficiently robust. However, specific concerns were consistently raised by DWOs as to consistent enforcement of the legislation, and the adequacy of financial penalties applied in cases involving animal cruelty, puppy farming and illegal activities. Given the profitability of puppy-farming in particular, fines of EUR 1000–2000 were perceived by the DWOs as insufficient sanctions which would not discourage re-offending. In particular, the DWOs reported poor compliance with microchipping legislation in regard to the dogs they received. Since the introduction of mandatory micro-chipping in 2015 [10], which resulted in an initial surge in uptake in 2016, an average of 87,787 dogs were microchipped per annum during 2017–2020 [3]. However, many DWOs reported cases in which dogs they rescued, or which were surrendered to them, were often without a microchip or the microchip displayed erroneous information. Poor linkage between the four licensed microchip companies in Ireland was reported as a significant barrier and has previously been identified as a concern in the UK [33]. There is limited data on compliance with microchipping rules in Ireland. High compliance has been reported elsewhere; for instance, in Rome, Italy, 75.3% of dogs were correctly registered and identified, with these animals more likely to be purebred, neutered, living in urban areas, and visiting a veterinarian frequently [34]. Microchipping/ tattooing (pre-2005) has been mandatory in Italy since 1991 [34].

A further prominent topic raised consistently by DWOs related to the management and regulation of dog breeding in Ireland; specifically, the operating practices of some DBEs/“puppy farms”. Participating DWOs claimed that some establishments engaged in inappropriate breeding activities (such as inbreeding or poorly matched crossbreeding), kept large numbers of dogs (up to 300 breeding bitches), and had poor sanitary standards. These observations were as reported by DWOs, and were not tested in the current study. In the UK, COVID-19 has been identified as a main contributor to an expansion of inappropriate and profit-garnering breeding [35], which is consistent with feelings expressed by DWOs in this study. In a previous study, More et al. (2022) identified 14,732 dog sales advertised in Ireland over a 6-month period: with an average price of EUR 830 and a maximum price of EUR 6500 [3]. The most common breeds advertised were poodle crosses, German Shepherds, and Golden Retrievers [3]. As of July 2022, at least six DBEs in Ireland had permission to keep over 100 breeding bitches, with one organisation licensed to have 300 breeding bitches on their premises [35]. The need for sustainability in dog supply and demand was identified as an international challenge by the International Partnership for Dogs pre-pandemic, and the need for greater intervention in Ireland seems clear. DWOs see poorly run DBEs and unlicensed dog breeding as a significant challenge both for themselves and the veterinary profession.

DWOs were more likely to report higher levels of surrendered dogs as the focus groups and interviews continued over the study period (January to May 2022). This coincided both with the reduction in frequency of working from home and the time when dogs purchased early in the pandemic were reaching sexual maturity (6–24 months old). It is at this age that behavioural issues stemming from poor training become more apparent [36]. The increased surrendering of dogs was identified as a future challenge for DWOs, many of which are already working at full capacity.

In terms of dog rehoming operations, many of the organisations interviewed described rigorous checks and policies, including home checks (virtual and in-person), and the matching of dogs to potential owners’ lifestyles. However, these practices were not universally adopted by DWOs. Similar to veterinary practices [37], COVID-19 led to a re-organisation of practices and procedures for DWOs, especially in relation to appointments and re-homing procedures. Organisations that incorporated the use of technology for home checks (such as video calls), and the introduction of mandatory appointments (rather than unannounced visits) for prospective owners during COVID-19, intended to continue these procedures post-pandemic. The People’s Dispensary for Sick Animals Animal Wellbeing (PAW) report in the UK, highlighted how the preparatory work of rescue organisations (and DBEs), such as utilising pre-purchase consultations and matching breeds to owner lifestyles, improved outcomes in rehoming [6]. This corresponds to our findings. Behavioural problems are more common among shelter dogs, and undesirable dog behaviours increased an animal’s chances of being returned to its source of purchase [38].

Some of the larger organisations with greater facilities and volunteer bases expressed a desire for increased inspection of facilities by authorised officers/local authorities, as a method to improve overall standards in the sector. In terms of next steps, some organisations suggested the development and implementation of standards similar to those from the ADCH in the UK. ADCH standards have minimum requirements for organisations, including disease control measures, behaviour modification techniques and appropriate facilities [39]; which some Irish DWOs believe would achieve better outcomes and improve dog welfare if implemented in Ireland. However, introduction of additional standards is likely to contribute to greater administrative burden and cost and could potentially reduce the viability of smaller rural rescues.

One unexpected finding was the position of greyhounds in the welfare ecosystem. Greyhound racing is a significant industry in Ireland, with the governing body (Greyhound Racing Ireland) receiving EUR 16.8m in governmental supports in 2019 [40]. By comparison, over the 5 years between 2016–2020, DAFM awarded EUR 13.9m to animal welfare organisations. DWOs highlighted greyhounds (and other sighthounds such as lurchers) as amongst the most difficult to rehome in Ireland, leading to many of these dogs being exported (according to the welfare organisations). Whilst DWOs indicated improving rehoming for sighthounds in Ireland, more work is needed in this area. Of note, New Zealand has adopted a tri-pronged approach of “reducing the number and increasing the ‘quality’ of greyhounds born in and/or imported into New Zealand, increasing racing opportunities to extend racing ‘careers’, and expanding rehoming opportunities” to replace the image of an athlete and predator into “an ‘unskilled’ companion-in-waiting” [41]. However, Stevens et al. stressed that the issues faced in the New Zealand greyhound industry are more systematic than simply rehoming issues, in agreement with our findings in the current study [41].

In terms of other challenges, many DWOs highlighted the high costs of veterinary care, coupled with their inability to host fundraisers during COVID-19, as threats to their viability. In addition, Brexit and the increased difficulties and paperwork in exporting dogs to and through the UK (predominantly to Scandinavia) creates further difficulties for the rehoming of dogs.

For recommendations from the findings, greater collaboration between Irish government and DWOs to co-develop and improve general practices across the board. Many DWOs highlighted the need for greater discussions with the government, Gardaí (police) and other DWOs, which respondents believed would improve dog welfare. Initiatives such as greater co-operation over cruelty cases/allegations, educational forums, such as an annual conference for all stakeholders, and greater increased collaboration between DWOs were highlighted. The DWOs indicated that current collaboration between DWOs is predominantly undertaken via informal connections or communications in relation to home checks. Additionally, research into the welfare of greyhounds and cats (noting that cats are frequently mentioned in the transcripts) are a logical next step to inform future animal welfare policies.

This novel research is the first study to investigate the perspectives of DWOs in relation to dog welfare in Ireland, and utilises a challenges- and solutions- paradigm using a rigorous academic approach in relation to dog welfare policies. This research directly involved DAFM but remained independent and confidential for participants to freely share their views. All organisations funded by DAFM in 2019–2021 were offered the opportunity to participate, in addition to some snowball recruitment in non-government funded DWOs to provide balance. The lack of research into general dog ownership and welfare in Ireland is a challenge in developing policy and recommendations for the future. This study is an initial step in identifying solutions to inform future policy development. It is recommended that future research in Irish dog welfare should consider the concerns raised in this study.

## 5. Conclusions

In conclusion, saturation of the dog market due to increased numbers of households acquiring dogs during COVID-19, limited understanding, and awareness in relation to dog welfare and husbandry, and inconsistent enforcement of current legislation were the key challenges identified by participating DWOs in relation to Irish dog welfare over the next five years. Additionally, the DWOs reported that finance, volunteer burden and a shortage of trained and/or accredited animal welfare professionals (veterinary behaviour specialists/trainers and veterinary surgeons/nurses) are likely to amplify these challenges in dog welfare. The DWOs believe that steps can be taken in terms of public education campaigns, improved enforcement and greater communication to help in reducing the impact of these challenges.

## Figures and Tables

**Figure 1 animals-12-03289-f001:**
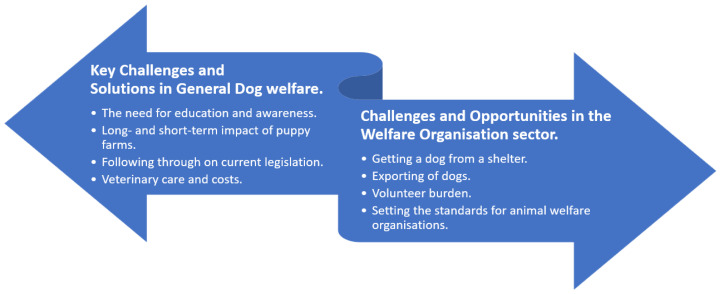
Thematic map diagram reflecting the opinions and beliefs of participating dog welfare organisations.

**Table 1 animals-12-03289-t001:** Demographic details of included Dog Welfare organisations.

Organisation Number	Focus Group or Interview	Size
1	Interview (I1)	Small
2 *	Interview (I2)	Medium
3 *	Interview (I3)	Medium
4	Interview (I4)	Medium
5	Interview (I5)	Small
6	Interview (I6)	Large
7	Interview (I7)	Small
8	Interview (I8)	Medium
9	Interview (I9)	Large
10	Interview (I10)	Small
11	Interview (I11)	Large
12	Interview (I12)	Large
13	Interview (I13)	Medium
14	Interview (I14)	Small
15 *	Interview (I15)	Medium
16	Interview (I16)	Medium
17	Focus Group (FG1.1)	Small
18	Focus Group (FG1.2)	Small
19	Focus Group (FG1.3)	Large
20	Focus Group (FG1.4)	Medium
21	Focus Group (FG2.1)	Large
22	Focus Group (FG2.2)	Large
23	Focus Group (FG2.3)	Medium
24	Focus Group (FG2.4)	Medium
25	Focus Group (FG3.1)	Small
26	Focus Group (FG3.2)	Small
27	Focus Group (FG3.3)	Small

* Had two participants in the interview. Small (10), Medium (10), Large (7).

## Data Availability

Restrictions apply to the availability of these data and is not available due to GDPR and privacy conserns.

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
