# Peer review of "A Qualitative Exploration of Challenges and Opportunities for Dog Welfare in Ireland Post COVID-19, as Perceived by Dog Welfare Organisations"

_animals, 2022, doi:10.3390/ani12233289_

Round 1
Reviewer 1 Report
Thank you for the opportunity to review this manuscript. The paper is timely, and insightful, and has a range of practical implications and so makes an important contribution to the field. I have some minor suggestions to improve the manuscript outlined below;
1) Suggest proofreading the simple summary as it is a little repetitive, and suggest changing the word 'intimately'
2) The introduction could do more to highlight the need for this research in Ireland and to discuss whether this research has been carried out in other countries and whether the authors feel the findings will be different or similar to other countries and why. Could strengthen the rationale for exploring these issues through the DWO's directly.
3) The number of DWO's that participated is reported, but there is no information available on how many participants from each DWO participated which is important information to include (e.g. did more participants come from a single organisation? This could mean the findings mostly reflected views of larger organizations for example). Highly recommend including a table that demonstrates demographics relating to participants themselves and a brief anonymous description of the organization that they represent. Each should have a clear participant number that can then be linked back to any direct quotes included from them in the results to give context to the view point/quote included.
4) It was mentioned that one male conducted the focus groups, but was this the same person for the interviews? Or did multiple people conduct the interviews? Need to be clearer.
5) Minor typos and need for proof reading, check for another repetitive language (e.g. lines 114 and 153). There is inconcsitency in the formatting of the Figure 1 (see capital letters)
6) Usually you have key themes and subthemes for TA yet only two key themes are presented and the subthemes have not been labelled subthemes. Suggest labelling these appropiately (e.g., Theme A, subtheme A1, A2, A3 etc.). This would help the reader follow the results more easily too.
7) Although qual analysis has been carried out, it would still be useful to think about including (either in text or a table) the data coverage for each theme and subtheme. For example, how many DWO's mentioned/discussed each challenge/solution, this could give indication to which perhaps are the most important (e.g., if all DWO's disucssed a signle issue than it's perhaps more of a prominent issue than something only mentioned by one DWO).
8) Quotes appear to be placed at the end of each paragraph rather than being integrated and interpreted throughout the analysis, and it is unclear whether some information (e.g., 3.1.2) was data-driven, or whether the authors are just citing research out side of quotes here.
9) Line 202 'up to 15 years' - lots of dogs live past 15 years?
10) Section 3.1.5 seemed less important, was this discussed less / by fewer DWO's?
11) Quotes should be linked directly to a particular participant from a particular DWO, rather than just saying which focus group/interview - need context for the quotes (also see point 3)
Author Response
Reviewer 1
Suggest proofreading the simple summary as it is a little repetitive, and suggest changing the word 'intimately'
Thank you, We have reduced some of the repetition in the summary and made some proof-reading changes to improve the flow. We have also changed the word intimately.
The introduction could do more to highlight the need for this research in Ireland and to discuss whether this research has been carried out in other countries and whether the authors feel the findings will be different or similar to other countries and why. Could strengthen the rationale for exploring these issues through the DWO's directly.
We have included more of the literature from outside Ireland and been more clear on the need for this research in an Irish context and how this addresses this as part of the larger research programme.
3) The number of DWO's that participated is reported, but there is no information available on how many participants from each DWO participated which is important information to include (e.g. did more participants come from a single organisation? This could mean the findings mostly reflected views of larger organizations for example). Highly recommend including a table that demonstrates demographics relating to participants themselves and a brief anonymous description of the organization that they represent. Each should have a clear participant number that can then be linked back to any direct quotes included from them in the results to give context to the view point/quote included.
As per line 155, 3 interviews were triadic; with the interviewer and 2 representatives from the organisation on the same call. This was due to the organisational representative not feeling comfortable in representing all the levels of their work on their own (such as the administration side and the care side). The actual size of the organisation did not play a part in having 2 representative in the interview. Additionally, weight was given per organisation rather than amount of participants and no organisation in the focus groups was represented by 2 individuals. We have now included in Table 1 which organisations had 2 representatives and included more information as per above to highlight the rationale behind triadic for some organisations (Line 155).
Secondly, confidentially of involvement was included as part of the agreement to contribute. The small nature of the population (ie there are only 74 organisations which could be included) raises issues of ease of identification. Only the 1st author and interviewer (BM), and the secondary analyst (CMcK) were aware of the organisations involved. However, we have included a table with some demographic information, Table 1; which makes clear which interviews were triadic and the size of the organisations.
4) It was mentioned that one male conducted the focus groups, but was this the same person for the interviews? Or did multiple people conduct the interviews? Need to be clearer.
Yes, it was the same interviewer for all focus groups and interviews. This has been clarified in section 2.1, Line 169.
5) Minor typos and need for proof reading, check for another repetitive language (e.g. lines 114 and 153). There is inconsistency in the formatting of the Figure 1 (see capital letters).
The piece has been proof-read thoroughly and typos corrected. Figure 1 has been reworked as a result of Reviewer 2’s feedback and formatting has been fixed.
6) Usually you have key themes and subthemes for TA yet only two key themes are presented and the subthemes have not been labelled subthemes. Suggest labelling these appropriately (e.g., Theme A, subtheme A1, A2, A3 etc.). This would help the reader follow the results more easily too.
The terminology used has been altered to key themes and subthemes.
7) Although qual analysis has been carried out, it would still be useful to think about including (either in text or a table) the data coverage for each theme and subtheme. For example, how many DWO's mentioned/discussed each challenge/solution, this could give indication to which perhaps are the most important (e.g., if all DWO's discussed a single issue than it's perhaps more of a prominent issue than something only mentioned by one DWO).
I disagree with the inclusion of data coverage per theme. I think that is a quantification of the work, which was not the intention. Another arm of the project has that aim. However, all themes and subthemes were discussed and were prominent issues. The most prominent (education, puppy farm, enforcement of current legislation and volunteer burden) are highlighted in each subtheme as particularly prominent. I have added into these themes that they were particularly prominent.
8) Quotes appear to be placed at the end of each paragraph rather than being integrated and interpreted throughout the analysis, and it is unclear whether some information (e.g., 3.1.2) was data-driven, or whether the authors are just citing research outside of quotes here.
Thank you, we have moved quotes to be closer to their interpretations. Due to the length of most of the quotations used, we feel they would create barriers to readability if totally within text.
On the research inclusion, some of the statements made by DWOs are contentious and we feel that referencing to the legal and statutory changes are helpful for the reader to understand the context, especially in relation to 3.2.2.
9) Line 202 'up to 15 years' - lots of dogs live past 15 years?
This line was to provide a general timeframe rather than a precise estimate. This has been altered to read (approximately 10-20 years) as a result of your feedback.
10) Section 3.1.5 seemed less important, was this discussed less / by fewer DWO's?
The sub-theme “Awareness through mass and social media” was discussed by the majority of organisations; however, the issues raised were very consistent and therefore the actual theme itself is smaller. However, with your feedback and delving into the analysis, we feel that this theme may be better situated within the “The need for education and awareness” subtheme and have integrated that into that theme instead.
11) Quotes should be linked directly to a particular participant from a particular DWO, rather than just saying which focus group/interview - need context for the quotes (also see point 3).
For the three interviews which were relevant for this, they have now been delineated as Interview (I2.1, I2.2): Interview (I3.1, I3.2); Interview (I15.1, I15.2) in Table 1.
Reviewer 2 Report
This is in a way a nice paper containing a qualitative study that is conducted in a decent way. However, it has one main shortcoming: It lacks a clear focus and thereby the ability to link the findings to other research literature in a meaningful way.
Here is how the research aim is described: "Dog welfare is a wide-ranging topic, with multiple challenges and varied legislation and practices. The research programme’s review study identified challenges in relation to dog sales, breeding, exports and imports and data quality for registration and identification of dogs [3]. Exploring these challenges and the roles of DWOs in the Irish welfare community formed the initial nucleus of our research aims." So, in essence the data gathered is not used to throw light on or more problems but instead we get a report about the data that could be used for this purpose. The many various issues presented are not really related in any other ways than by being about dogs in Ireland and about what concerns Irish DWO's. This may qualify for a report but not for a scientific paper.
Author Response
Dear Reviewer,
Thank you for taking the time to read our article and conduct the review. We acknowledge that there are issues with linkage to previous work in terms of the Irish dog welfare sector; and we have outlined the lack of previous research and even relevant data on dogs in Ireland. The review piece conducted as part of the research programme (More et al, 2021) has outlined the issues on this and the position of Irish dog welfare legislation and practice to international research. We have made alterations to the article that have improved the research linkage and base within the introduction; to better illustrate the need for this research. Similarly, we have improved the reporting of the methods significantly.
As highlighted within the Introduction and above, the lack of data and previous research led to this exploratory study on dog welfare in Ireland. Your statement that the issues are “not really related in any other ways than being about dogs in Ireland and about what concerns Irish DWOs”, we feel misses the aim of the study; which was to identify these challenges and proper solutions using a source of data (Irish DWOs) that have previously been ignored in the scientific literature. This lack of previous data meant that the inductive methodology was needed to identify the problems for future research and the development of impact in regards legislation and action from the results of this study.
We feel that this is vital in the next steps in developing the research in Ireland and provides an interesting and pertinent piece of timely research on dog welfare; with a strong scientific basis.
Reviewer 3 Report
Overall, this is a well-prepared and well-written paper based on novel research which helps to advance knowledge on an important and understudied topic: perceptions of challenges and opportunities in relation to dog welfare in Ireland post COVID-19, as perceived by representatives of Irish dog welfare organisations. The study’s findings will be of particular interest to both the animal welfare sector and those involved in policy making. Conducting a qualitative study to explore this topic was ideal, given the lack of existing research in this area. I recommend this paper be published following the addition of more information to clarify and justify their choice of method.
My main suggestion for how the paper could be strengthened relates to the choice and explanation of the methods used. The authors reportedly follow “inductive thematic analysis”, citing Braun and Clarke’s 2006 paper ‘Using Thematic Analysis in Psychology’. Thematic Analysis refers to a range of qualitative methods, including three broad approaches: reflexive thematic analysis, coding reliability approaches, and codebook approaches. Each approach has different processes and assumptions. The authors should more clearly define the specific type of TA they used, justifying this through consideration of the research aims. Since writing their 2006 paper, Braun and Clarke have published further work to clarify and revise some of their thinking and processes around thematic analysis. Familiarity with their more recent work might be helpful to consult (e.g. 2019, https://doi.org/10.1080/2159676X.2019.1628806 2016 https://doi.org/10.1080/13645579.2016.1195588 and 2022 https://doi.org/10.1037/qup0000196).
One important point of clarification that Braun and Clarke have made is around conceptualisations of “themes”. Within the different approaches to TA, themes are conceptualised in different ways. The themes presented in this paper appear to be summaries of what participants said about a particular topic or in response to a particular question (linked to this, it would be helpful to see the semi-structured schedule: it would be great if this was available as supplementary material). There does not appear to be an underlying concept that underpins what participants said about challenges and opportunities in the welfare organisation sector and ties the subthemes together. Instead, it is the topic (i.e. “Key challenges and solutions in dog welfare” or “Challenges and opportunities in the welfare organisation sector”) that appears to unite the themes rather than patterns of shared meanings. The themes therefore appear to reflect what Braun and Clarke describe as “topic summaries” or “domain summaries” (common in coding reliability or codebook approaches) rather than fully realised themes of shared meaning underpinned by a central concept (the product of a reflexive approach to TA. In 2018 Braun and Clarke noted that ‘In writing our 2006 paper, we again took for granted that most readers would understand what a (fully realised) theme is, and how this differs from a summary of participant responses to a particular data collection question, or in relation to a particular area or ‘domain’ of the data’ (https://doi.org/10.1002/capr.12165). As mentioned above, it is not clear if which approach the authors took or how they conceptualised “themes” and this needs clarification.
The type of thematic analysis used also has implications for conceptualisations of saturation (lines 148-9). See https://doi.org/10.1080/2159676X.2019.1704846
I really like visual representations of themes (e.g. thematic maps) but I’m not convinced that the thematic map included accurately represents the themes. The two overarching themes are overlapping in the map, but this overlap is not made explicit in the write up. Rather, the two overarching themes are identified as quite distinct. Therefore, I don’t think it makes sense for the themes to be displayed as overlapping. If this is not the case, then further explanation of their linkage should be included.
I also have some minor suggestions to help strengthen the paper.
Methods
As I read the Methods section I was hoping to see inclusion of when (i.e. year/months) the interviews/focus groups take place. I see this is included later in the Results section (line 247). I think this should be noted in the Methods section to help readers contextualise the research earlier on.
Line 144: were all the interviews also conducted by BM? Would be good to include for consistency of reporting.
Was the semi-structured schedule the same across the interview/focus groups? It would be great to include the questions as supplementary material.
Were any extra insights were gleamed from FGs that were not possible through interviews? For instance, through interaction between participants.
How was it decided which DWO participated in an interview and which in a FG? Did the DWO get to choose? This information should be included.
Were any steps were taken to preserve confidentiality or participant’s anonymity in the focus groups? This information should be included.
Were the interviews transcribed by the researcher/s involved or an external person/company? This information should be included.
Results
What do the double slashes indicate within the quotes? Does this mean data has been omitted within the excerpt? Could you clarify this perhaps alongside the information given in lines 162-3?
Do you have a reference for the estimated figure given on line 219?
Discussion
Line 635: “conference/s” is repeated – should one of these be a different word?
Author Response
Overall, this is a well-prepared and well-written paper based on novel research which helps to advance knowledge on an important and understudied topic: perceptions of challenges and opportunities in relation to dog welfare in Ireland post COVID-19, as perceived by representatives of Irish dog welfare organisations. The study’s findings will be of particular interest to both the animal welfare sector and those involved in policy making. Conducting a qualitative study to explore this topic was ideal, given the lack of existing research in this area. I recommend this paper be published following the addition of more information to clarify and justify their choice of method.
My main suggestion for how the paper could be strengthened relates to the choice and explanation of the methods used. The authors reportedly follow “inductive thematic analysis”, citing Braun and Clarke’s 2006 paper ‘Using Thematic Analysis in Psychology’. Thematic Analysis refers to a range of qualitative methods, including three broad approaches: reflexive thematic analysis, coding reliability approaches, and codebook approaches. Each approach has different processes and assumptions. The authors should more clearly define the specific type of TA they used, justifying this through consideration of the research aims. Since writing their 2006 paper, Braun and Clarke have published further work to clarify and revise some of their thinking and processes around thematic analysis. Familiarity with their more recent work might be helpful to consult (e.g. 2019, https://doi.org/10.1080/2159676X.2019.1628806 2016 https://doi.org/10.1080/13645579.2016.1195588 and 2022 https://doi.org/10.1037/qup0000196).
One important point of clarification that Braun and Clarke have made is around conceptualisations of “themes”. Within the different approaches to TA, themes are conceptualised in different ways. The themes presented in this paper appear to be summaries of what participants said about a particular topic or in response to a particular question (linked to this, it would be helpful to see the semi-structured schedule: it would be great if this was available as supplementary material). There does not appear to be an underlying concept that underpins what participants said about challenges and opportunities in the welfare organisation sector and ties the subthemes together. Instead, it is the topic (i.e. “Key challenges and solutions in dog welfare” or “Challenges and opportunities in the welfare organisation sector”) that appears to unite the themes rather than patterns of shared meanings. The themes therefore appear to reflect what Braun and Clarke describe as “topic summaries” or “domain summaries” (common in coding reliability or codebook approaches) rather than fully realised themes of shared meaning underpinned by a central concept (the product of a reflexive approach to TA. In 2018 Braun and Clarke noted that ‘In writing our 2006 paper, we again took for granted that most readers would understand what a (fully realised) theme is, and how this differs from a summary of participant responses to a particular data collection question, or in relation to a particular area or ‘domain’ of the data’ (https://doi.org/10.1002/capr.12165). As mentioned above, it is not clear if which approach the authors took or how they conceptualised “themes” and this needs clarification.
The type of thematic analysis used also has implications for conceptualisations of saturation (lines 148-9). See https://doi.org/10.1080/2159676X.2019.1704846
Thank you for this and for providing a really clear description. On this, we have included more detail on the methodology that we incorporated to develop the analysis, with the utilisation of the Nowell Criteria (doi:10.1177/1609406917733847.) We hope that this clears that part up. In terms of the key themes and subthemes that we have identified, we feel that these are realised to their current potential; given the lack of prior qualitative research in the area. By developing the themes in this way (separating them into challenges and solutions for general dog welfare, and in the welfare sector), we feel this will be more impactful in effecting policy change and enabling further research into the topics uncovered.
I really like visual representations of themes (e.g. thematic maps) but I’m not convinced that the thematic map included accurately represents the themes. The two overarching themes are overlapping in the map, but this overlap is not made explicit in the write up. Rather, the two overarching themes are identified as quite distinct. Therefore, I don’t think it makes sense for the themes to be displayed as overlapping. If this is not the case, then further explanation of their linkage should be included.
Upon reflection, we agree and have made changes to the figure. Whilst there are undoubtably close links between the key themes, we agree it is better to keep them as distinct as possible.
I also have some minor suggestions to help strengthen the paper.
Methods
As I read the Methods section I was hoping to see inclusion of when (i.e. year/months) the interviews/focus groups take place. I see this is included later in the Results section (line 247). I think this should be noted in the Methods section to help readers contextualise the research earlier on.
Thank you, this has now been included. Line 162. In the methods.
Line 144: were all the interviews also conducted by BM? Would be good to include for consistency of reporting.
Yes, it was the same interviewer for all focus groups and interviews. This has been clarified in section 2.1, Line 169.
Was the semi-structured schedule the same across the interview/focus groups? It would be great to include the questions as supplementary material.
Yes, the same. Now included in supplementary and highlighted in the methods, Line 164.
Were any extra insights were gleamed from FGs that were not possible through interviews? For instance, through interaction between participants.
The findings were really consistent between the methods; which is both a positive for the overall story and a negative for further discussion. We have included more detail on this on lines 208-209.
Results
How was it decided which DWO participated in an interview and which in a FG? Did the DWO get to choose? This information should be included.
Organisations were offered the choice of participation in a focus group or an individual interview. This is now clarified in line 138.
Were any steps were taken to preserve confidentiality or participant’s anonymity in the focus groups? This information should be included.
“Participants were informed that their participation was anonymous outside of their specific focus group, and that they would be interacting with organisations representatives openly within their focus group session. Anyone with any issues with this lack of anonymity was offered an individual interview. “ Now included in line 139-41.
Were the interviews transcribed by the researcher/s involved or an external person/company? This information should be included.
Transcription was conducted by an external agency and the transcripts were checked by BM and CMcK for precision. This is now included on line 171-172.
Results
What do the double slashes indicate within the quotes? Does this mean data has been omitted within the excerpt? Could you clarify this perhaps alongside the information given in lines 162-3?
// was to remove words from the quotes to improve readability. We have clarified that more fully in lines 195-196 for this.
Do you have a reference for the estimated figure given on line 219?
Apologies, We have checked line 219 and are not sure what you mean. The formatting may have changed the line number. If it is in relation to “some larger DWOs were able to conduct dog training classes as part of their rehoming policies; predominantly over Zoom/Skype”; we do not have a reference for the number. Alternatively, if it is in relation to “Mandatory dog ownership courses, with reference to German and Swiss practices, were seen as another possible solution by DWOs.” We have included a reference for these. Line 257
Discussion
Line 635: “conference/s” is repeated – should one of these be a different word?
Yes, it’s a typo we missed. Thank you.